//doi.org/10.1038/s41467-023-41509-x

# Genotypic diversity and unrecognized antifungal resistance among populations of *Candida glabrata* from positive blood cultures

Hassan Badrane [1], Shaoji Cheng[1], Christopher L. Dupont [2], Binghua Hao[1], Eileen Driscoll[1], Kristin Morder[1], Guojun Liu[1], Anthony Newbrough[1], Giuseppe Fleres[1], Drishti Kaul[2], Josh L. Espinoza[2], Cornelius J. Clancy[1,3,4] & M. Hong Nguyen [1,4] ✉

The longstanding model is that most bloodstream infections (BSIs) are caused by a single organism. We perform whole genome sequencing of five-to-ten strains from blood culture (BC) bottles in each of ten patients with *Candida glabrata* BSI. We demonstrate that BCs contain mixed populations of clonal but genetically diverse strains. Genetically distinct strains from two patients exhibit phenotypes that are potentially important during BSIs, including differences in susceptibility to antifungal agents and phagocytosis. In both patients, the clinical microbiology lab recovered a fluconazole-susceptible index strain, but we identify mixed fluconazole-susceptible and –resistant populations. Diversity in drug susceptibility is likely clinically relevant, as fluconazole-resistant strains were subsequently recovered by the clinical laboratory during persistent or relapsing infections. In one patient, unrecognized respiration-deficient small colony variants are fluconazole-resistant and significantly attenuated for virulence during murine candidiasis. Our data suggest a population-based model of *C. glabrata* genotypic and phenotypic diversity during BSIs.

*Candida* spp. are remarkably versatile opportunistic pathogens capable of colonizing or causing invasive infections of diverse anatomical sites. Candidemia is the most common type of invasive candidiasis and the fourth leading bloodstream infection (BSI) in the United States[1,2]. *Candida glabrata*, the second most prevalent agent of invasive candidiasis, is notable among *Candida* spp. for its haploid, rather than diploid genome, and its propensity to develop antifungal resistance. Like most *Candida*, *C. glabrata* are human gastrointestinal (GI) tract commensals. In the face of disturbed homeostasis, as in persons who are critically ill, immunosuppressed, receiving broad spectrum antibiotics, undergoing GI surgery or suffering from disruptions of

mucosal integrity, *C. glabrata* can invade the bloodstream and other deep tissues. *Candida* strains often manifest striking genomic plasticity, which is believed to facilitate adaptation and survival under diverse and changing conditions[3–10]. Whole genome sequencing (WGSing) of longitudinal *Candida* strains from humans or infected mice often reveal substantial genomic differences, including point mutations, insertions, deletions, and whole or partial chromosomal aneuploidies[7].

The longstanding model is that most BSIs, including candidemia, are caused by a population of genetically identical strains derived from a single organism that passes through a bottleneck ("single organism"

[1]University of Pittsburgh, Pittsburgh, PA, USA. [2]J. Craig Venter Institute, La Jolla, CA 92037, USA. [3]VA Pittsburgh Healthcare System, Pittsburgh, PA, USA. [4]These authors contributed equally: Cornelius J. Clancy, M. Hong Nguyen. ✉e-mail: mhn5@pitt.edu

or "independent action" hypothesis)[11,12]. Increasingly, WGS data demonstrate that colonization or chronic infections by various bacteria may be caused by a population of strains in which genetic diversity emerges during long-term host interactions[13–15]. At present, it is unknown if BSIs are commonly caused by genetically diverse populations, rather than single organisms. We recently demonstrated that positive blood cultures from patients with monomicrobial bacterial BSI (carbapenem-resistant *Klebsiella pneumoniae*) were comprised of mixed populations of genetically and phenotypically diverse strains, including those demonstrating differences in antibiotic susceptibility and virulence[16].

In this study, we tested the hypothesis that contemporaneous *C. glabrata* strains from individual patients with BSIs were genetically and phenotypically diverse. We evaluated *C. glabrata* BSIs in 10 patients. For each patient, the clinical lab isolated an index strain from a single colony morphotype from an initial positive blood culture bottle. We analyzed WGSs of the index and 4-9 other strains recovered from independent colonies. We then assessed phenotypes of genetically distinct strains from each of 2 patients. As hypothesized, we detected within-patient genotypic and phenotypic diversity, including antifungal resistance, that was not recognized by the clinical laboratory.

## Results

### Patients with *C. glabrata* BSIs

We enrolled 10 patients with *C. glabrata* BSIs (6 men and 4 women). Patients' age ranged from 27 to 81 years (median age: 59.5 years). Nine patients had underlying diseases of the GI or biliary tracts. Two patients had breakthrough *C. glabrata* BSIs while receiving fluconazole, and 3 others received an antifungal agent in the 3 months preceding BSI (fluconazole or caspofungin). Four patients had deep-seated invasive candidiasis (peritonitis, empyema, cholangitis) in addition to candidemia, and 4 had concomitant bacterial infections. In one patient, the index *C. glabrata* strain (i.e., strain isolated by the clinical microbiology laboratory) was fluconazole-resistant; index strains from the other 9 patients were fluconazole-susceptible-dose dependent. In all, 1 patient had relapsing invasive candidiasis 47 days after the initial episode of candidemia, and 3 patients died within 30 days of candidemia.

### WGS of *C. glabrata* strains

We streaked 10 μL aliquots from the initial positive blood culture bottle from each of the 10 patients onto Sabouraud dextrose agar (SDA) plates, which were incubated at 35 °C for 48 h. For 9 patients, colonies were indistinguishable by morphotypes. We isolated strains from 4 to 9 randomly selected colonies. In the tenth patient (patient J),

colonies were indistinguishable by morphotypes at 48 h of incubation. At 84 h, however, pinpoint colonies were evident, admixed with a greater number of larger colonies (Fig. 1). For the remainder of the manuscript, we refer to the larger and pinpoint colonies as normal (NCV) and small colony variants (SCV), respectively. We isolated 4 NCV and 5 SCV strains for further study. The index J strain grew as a NCV. For each patient, the index strain was labeled as strain #1, and other independently isolated strains were labeled as #2 up to #10. In patient J, strains #1-5 and #6-10 were NCV and SCV, respectively, on SDA plates. All BSI strains underwent next generation sequencing (Illumina Next-Seq). Index strains also underwent sequencing by Oxford Nanopore (ONT) using MinION.

Strains had 13 chromosomes and closed genomes of 12.7–12.98 Mbp. The closed genome of reference strain *C. glabrata* CBS138 was 12.34 Mbp. On average, 98% of reads from clinical *C. glabrata* were mapped to the reference genome (range: 96–99%). Single nucleotide polymorphisms (SNPs) and insertions-deletions (indels) within-patient were not distributed uniformly along chromosomes, but rather were predominantly clustered in hot spots, particularly in subtelomeric regions (Supplementary Fig. S1).

### Genotypic diversity of *C. glabrata* strains

To estimate genetic relationships among the 94 *C. glabrata* strains from positive blood cultures, we built a high-resolution phylogenetic tree based on nucleotide differences (SNPs or indels), estimated by comparison with *C. glabrata* CBS138 (Fig. 2a and Supplementary Fig. S2). Overall, SNPs and indels accounted for 89% and 11% of nucleotide differences, respectively. Strains represented 4 sequence types (STs), and they segregated into three distinct clades: (1) ST3 (patients J, K, P, ST), (2) ST26 (patient H), (3) ST10 (patients C, D, I, L), and ST179 (patient EF). Within clades, strains clustered by patient. Nodes grouping within-clade and within-patient strains had high bootstrap support values (≥85). Deduced nucleotide sequence identity ranged from 99.02% to 99.98% for pairwise comparison of all clinical isolates (Supplementary Table S1). Despite this genetic relatedness, within-patient strains were clearly divergent, based on unique SNPs and/or indels. There was a within-patient average of 3,598 nucleotide differences. J and C strains showed the highest (7,507) and lowest (2,067) within-patient nucleotide differences, respectively.

Results were similar when strains were compared by number of genome variants (i.e., unique genome sites at which SNPs or indels were observed). Representative Venn diagrams for within-patient genome variant comparisons are shown in Fig. 2b. On average, 70.4% and 29.6% of variants were synonymous and non-synonymous, respectively; 99.5% and 0.5% of non-synonymous SNPs were missense

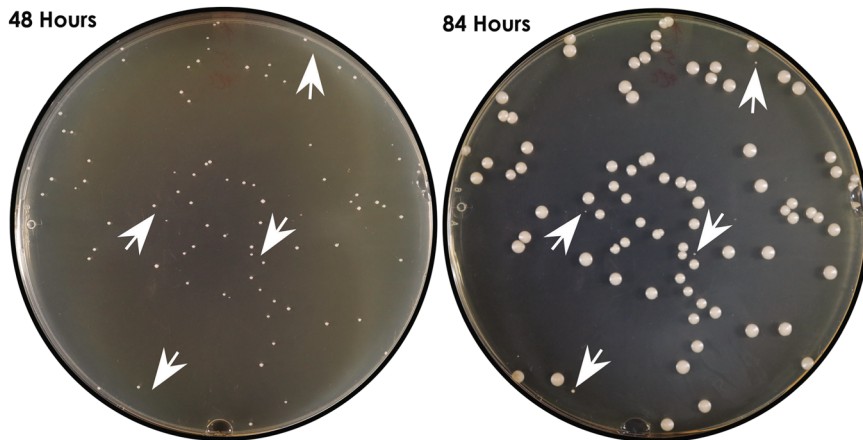

**Fig. 1 | Morphology of *C. glabrata* strains recovered from the positive blood culture of patient J.** Ten microliters from the blood culture bottle were plated onto Sabouraud dextrose agar plates and incubated at 35° C. White arrow heads point to colonies which are evident at 84 h (right panel) but not at 48 h (left panel).

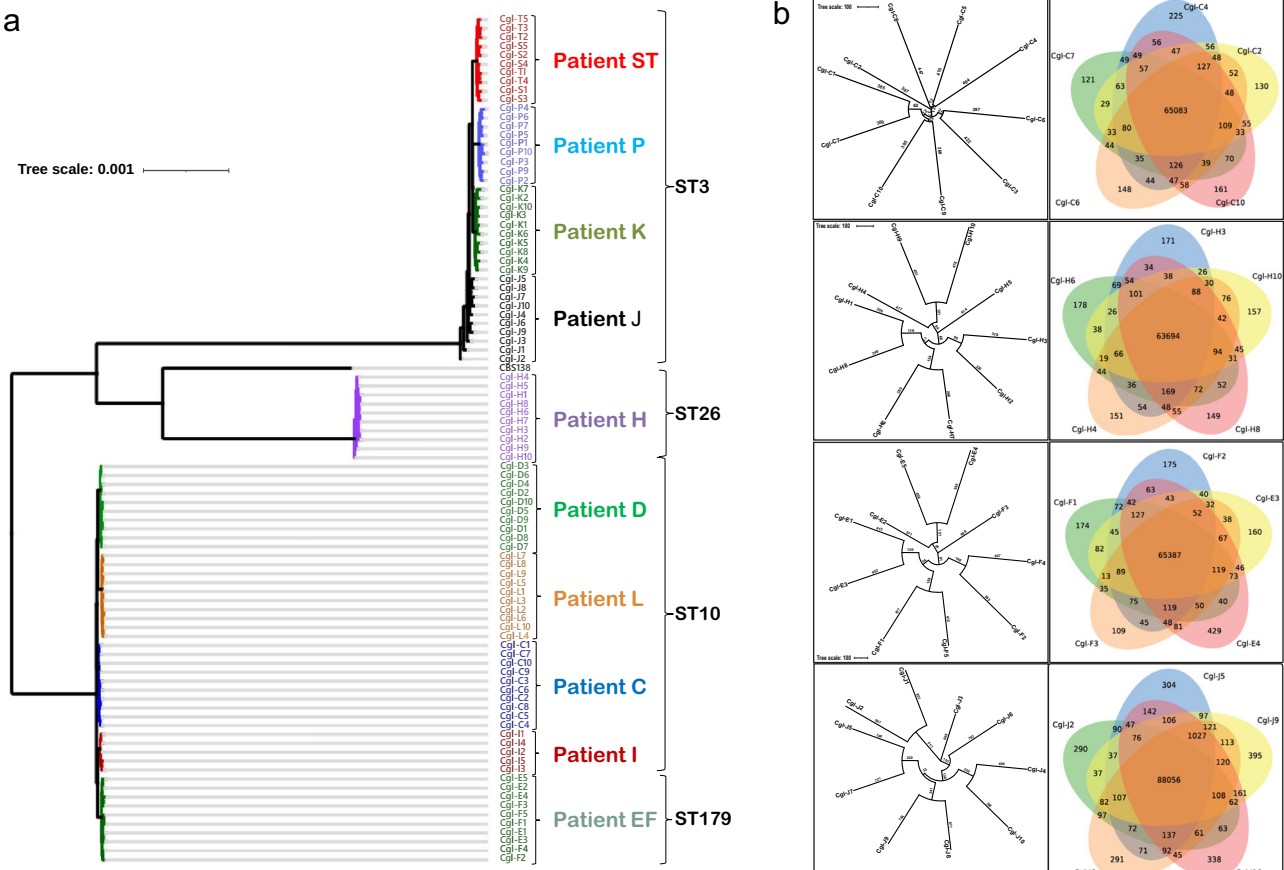

**Fig. 2 | Phylogenomic analysis of *C. glabrata* strains from blood cultures of patients with bloodstream infections. a** Unrooted phylogenomic tree of the 94 BSI strains and reference strain CBS138, as estimated using Maximum Likelihood with RaxML. It is based on a genome alignment generated from variant calling data. Within-patient diversity is demonstrated for all strains, in each patient. Strains from a given patient are color coded and sequence type is shown to the right. **b** Unrooted phylogenomic radial tree built with parsimony using PAUP is shown for each of 10 strains from 4 representative patients. Each patient was infected with a different *C. glabrata* ST. Also shown are Venn diagrams of numbers of genome variants, comparing 5 *C. glabrata* strains from individual patients.

and nonsense, respectively. We analyzed variants associated with non-synonymous mutations that discriminated within-patient strains (i.e., mutations found in at least one strain from a patient but not found in all strains, compared to *C. glabrata* CBS138). Gene Ontology (GO) term analysis of these variants revealed a significant over-representation of genes encoding adhesins and cell wall proteins (Supplementary Table S2). J strains were also significantly enriched for non-synonymous variants in genes involved in mitochondrial functions.

We identified within-patient gene copy number variants (CNVs) in 4 patients (patients EF, J, H, and K) using a genome assembly and alignment approach with a Nextflow pipeline. Twelve genes were deleted among strains from these patients, 7 of which encoded adhesins (Table 1). There were no within-patient gene duplications. To look for structural variants, genomes of within-patient strains were aligned using progressiveMauve, then visualized in Mauve Alignment Viewer. Graphics did not reveal aneuploidies, but provided evidence of chromosomal rearrangements (Supplementary Fig. S3a). We used similarity dot plots of pairwise genome comparisons (D-Genies) to confirm chromosomal rearrangements in strains from 8 patients (patients K and ST excepted) (Supplementary Fig. S3b).

We constructed a phylogenetic tree using maximum likelihood based on nucleotide differences in mitochondrial genomes among the 94 *C. glabrata* BSI strains (Fig. 3a). Mitochondrial genome homology (i.e., DNA identity) among strains from patient J was 96.71–99.98% (Supplementary Table S1). For other strains, within-patient homology was higher (99.69–100% DNA identity). Overall, the 94 *C. glabrata* strains were segregated by mitochondrial

phylogeny into the same clades as with nuclear phylogeny. H strains again constituted their own clade. Within the other 2 clades, strains did not cluster cleanly by patient due to very high DNA identity. Strains J1-J5 carried fewer mitochondrial nucleotide differences than did J6-J10 (Fig. 3a, b), and they clustered in a clade with P, K and ST strains, in keeping with nuclear phylogeny. Strains J6-J10 showed much higher divergence (DNA identity: 96.71–99.21%) than did J1-J5 (99.71–99.98%).

In summary, we demonstrated genotypic diversity among *C. glabrata* strains recovered from positive blood cultures in each of 10 patients. We next evaluated whether certain genotypically distinct strains manifested differences in clinically relevant phenotypes. We studied strains from patients L and J in greater detail because genomic data suggested within-patient differences that might be associated with antifungal resistance that was not recognized by the Clinical Microbiology Laboratory.

## Phenotypic diversity of *C. glabrata* strains from patient L

Upon diagnosis of fluconazole-susceptible *C. glabrata* BSI, patient L was treated with IV fluconazole. Blood cultures ~48 h later remained positive for *C. glabrata*, which was now identified by the clinical microbiology laboratory as azole-resistant.

Strains L1-L10 were similar in colony size and morphology on YPD and RPMI agar plates, and in growth rates in YPD broth at 37 °C. Genome variants that discriminated between L strains were noted in 53 genes encoding adhesins (Supplementary Data 1) We assessed adherence of representative strains L2, L4, and L6 to Hep-2 cells in vitro.

**Table 1 | Within-patient gene copy number variants**

| Gene | Strains | | | | | | | | | | Protein encoded by specific gene |
|---|---|---|---|---|---|---|---|---|---|---|---|
| **Patient EF strains** | E1 | E2 | E3 | E4 | E5 | F1 | F2 | F3 | F4 | F5 | |
| *EPA14* | ■ | ■ | □ | □ | ■ | ■ | ■ | □ | ■ | □ | Putative cell wall protein (adhesin cluster I) |
| *TNR1* | ■ | ■ | ■ | □ | □ | ■ | ■ | ■ | ■ | □ | Putative nicotinamide transporter |
| **Patient EF strains** | J1 | J2 | J3 | J4 | J5 | J6 | J7 | J8 | J9 | J10 | |
| *EPA1* | □ | □ | □ | □ | ■ | □ | □ | □ | □ | ■ | Putative cell wall protein (adhesin cluster I) |
| *EPA5* | □ | □ | □ | ■ | □ | ■ | □ | □ | ■ | □ | Putative cell wall protein (adhesin cluster I) |
| CAGL0E00187g | ■ | □ | □ | □ | □ | □ | □ | ■ | □ | ■ | Putative adhesin-like protein (adhesin cluster IV) |
| *rRNA-18S* | ■ | ■ | ■ | ■ | ■ | ■ | ■ | ■ | □ | ■ | rRNA-18S ribosomal RNA |
| **Patient H strains** | H1 | H2 | H3 | H4 | H5 | H6 | H7 | H8 | H9 | H10 | |
| *RDN25-1* | ■ | ■ | ■ | ■ | ■ | ■ | ■ | □ | ■ | ■ | 25S rRNA |
| *RDN18-1* | ■ | ■ | ■ | ■ | ■ | ■ | ■ | □ | ■ | ■ | 18S rRNA |
| CAGL0A04851g | ■ | □ | ■ | ■ | ■ | ■ | ■ | □ | ■ | ■ | Pseudogene, Putative cell wall protein (adhesin cluster III) |
| CAGL0E00231g | □ | □ | □ | ■ | ■ | ■ | ■ | □ | □ | □ | Putative cell wall protein (adhesin cluster III) |
| CAGL0G00099g | □ | □ | □ | ■ | ■ | □ | □ | □ | □ | □ | Putative cell wall protein (adhesin cluster III) |
| **Patient K strains** | K1 | K2 | K3 | K4 | K5 | K6 | K7 | K8 | K9 | K10 | |
| *RDN18-1* | □ | □ | ■ | ■ | ■ | ■ | ■ | ■ | □ | ■ | 18s rRNA |

Data were analyzed using the Nextflow pipeline. Black and white boxes represent presence and absence of specific genes, respectively. Deletions of 12 genes conferred within-patient differences between strains in 4 patients; 7 of these genes encoded adhesins. We did not observe any within-patient gene duplications.

These strains harbored mutations in 22, 30, and 30 adhesin genes, respectively. Three, eight, and five adhesin genes had mutations in the respective strains, without being mutated in the other two strains. Adherence of strain L2 to Hep-2 cells was significantly higher (mean ± standard error of mean (SEM) 91.6 ± 1.7%) than that of either L4 (66.3 ± 0.8%, mean difference of 25.3% (95% CI: 21.0–29.5%), $p < 0.0001$) or L6 (78.4 ± 1.0%, mean difference of 13.2% (95% CI: 8.9–17.4%), $p < 0.0001$; one-way ANOVA with Dunnett's multiple comparisons test) (Supplementary Table S1).

Fluconazole, posaconazole and isavuconazole MICs against strains L4 and L8 were ≥4-fold, ≥16-fold, and ≥32-fold higher, respectively, than those against other L strains (Fig. 4). L4 carried a mutation within *PDR1*, conferring a G346C substitution within the Pdr1 central regulatory domain[17]. Pdr1 regulates transcription of multidrug transporter gene *CDR1*, which is linked to fluconazole resistance[18]. Using RT-PCR, we showed that *PDR1* and *CDR1* expression were significantly higher in L4 than in other L strains (Fig. 4). We deleted *PDR1* in strains L4 and BG2, then engineered either a *PDR1* G346C mutation or wild-type *PDR1* in these backgrounds. Azole MICs were higher against engineered *PDR1* G346C strains (e.g., 256 and 64 μg/mL fluconazole against engineered L4 and BG2, respectively) than against respective wild-type *PDR1* strains (32 and 16 μg/mL fluconazole, respectively) (Supplementary Table S3). L8 harbored wild-type *PDR1*; its *CDR1* expression was >11-fold greater than that of index strain L1 (Fig. 4). Of note, azole-resistant *C. glabrata* strains recovered from the patient's blood culture obtained after ~48 h of fluconazole treatment also harbored *PDR1* G346C.

### Phenotypic diversity of *C. glabrata* strains from patient J

Patient J was a liver transplant recipient who was treated with caspofungin for *C. glabrata* BSI. After completing treatment, he was placed on voriconazole prophylaxis, in keeping with our transplant program protocol[19]. Forty-seven days after BSI, he was

diagnosed with intra-abdominal infection due to fluconazole-resistant *C. glabrata* SCV (strain J11).

Strains J1-J4 were NCVs in size and morphology on YPD and RPMI agar plates, and in growth rates in YPD broth at 37 °C. Strains J6-J10 and J11 were consistent with SCVs (i.e., respiration-deficient *C. glabrata* petite mutants by a constellation of phenotypes, including defective growth in YPD liquid medium, inability to grow on YP-glycerol medium, and deep-violet color on agar containing eosin Y and trypan blue) (Fig. 5a, b and Supplementary Table S1)[20]. Strain J5, which grew as NCV at 48 h on SDA agar, was intermediate to strains J6-J10 and J1-J4 in growth rate in liquid YPD medium. By flow cytometry, cells of strain J5 were comparable in size to those of strains J6-J10 following growth in liquid YPD, and significantly smaller than those of strains J1-J4 (Fig. 5c). J5 also resembled SCV strains J6-J10 in inability to grow on YP-glycerol medium, and in deep-violet color on eosin Y and trypan blue indicator plates. Therefore, on balance, strain J5 was most consistent with a SCV.

We used GENOME-STRip to estimate mitochondrial genome copy numbers (mtN) of J strains from mapped short reads. Strains J1–J4 and J6 were estimated to have 30 and 35 mtN, respectively (Supplementary Table S1). Strains J5 and J7–J10 were estimated to have 1-18 mtN. We further estimated mtN of NCV strain J1 and SCV strains J9 and J11 by quantitative PCR. Mean mtN was significantly higher for J1 than for J9 or J11 (18.4 ± 2.9 vs 0.4 ± 0.1 and 0.3 ± 0.1, respectively; $p = 0.001$, one-way ANOVA). J1 also had significantly greater mitochondrial staining with rhodamine-123, as evident by FACS sorting, than did J9 (Fig. 6a). Electron microscopic images of J1 and J9 corroborated striking differences in mtN per cell (Fig. 6b).

To assess respiration in greater detail, we exposed strains J1 and J9 to various mitochondrial electron transport chain inhibitors. As expected, oxygen consumption by strain J1 was significantly reduced upon exposure to antimycin A, sodium cyanide or salicylhydroxamic acid (SHAM) (Fig. 6c). In contrast, oxygen consumption by strain J9 was already diminished in the absence of electron transport inhibition,

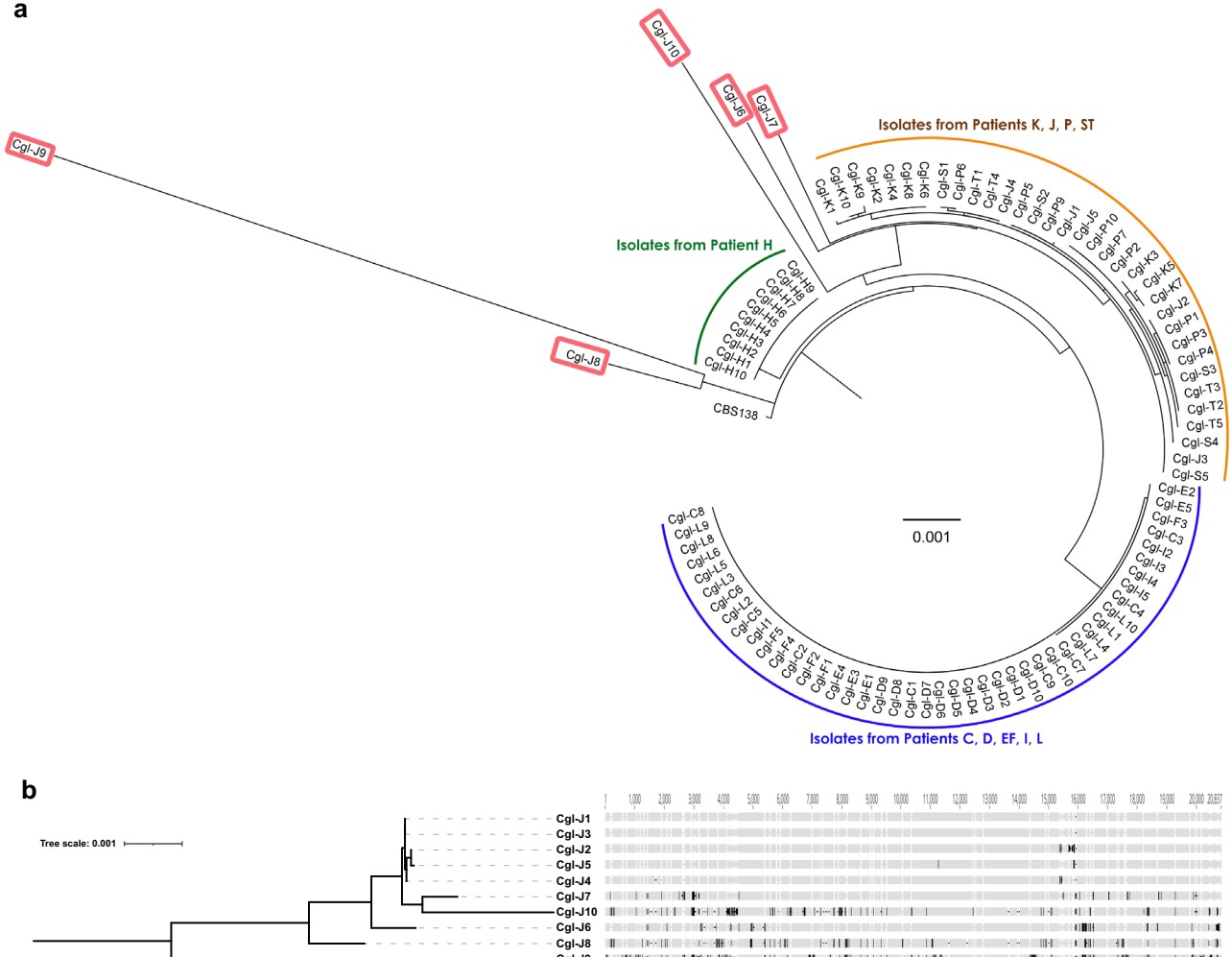

**Fig. 3 | Mitochondrial genome phylogeny of *C. glabrata* strains from blood cultures of patients with bloodstream infections. a** Unrooted phylogenetic tree for all 94 strains, estimated using Maximum Likelihood with RaxML. Strains J1-J5 carried fewer mitochondrial nucleotide differences than did J6-J10, and they clustered in a clade with K, P and ST strains, in keeping with nuclear phylogeny. Strains J6-J10 (red rectangle) showed much higher divergence than did J1-J5, and fall outside of the clade that include strains J1-J5. **b** High resolution mitochondrial genome phylogeny of 10 *C. glabrata* strains from patient J. Alignment graph of mitochondrial genomes from the 10 patient strains (right), and corresponding unrooted phylogenetic tree (left). Coordinates of nucleotide position are shown on top of alignment. Gray colored positions represent nucleotides identical to the consensus. Black vertical bars show presence of SNPs or insertions, and black horizontal lines show deletions. Phylogeny was estimated using RaxML.

and it was not reduced upon exposure to antimycin A, sodium cyanide or SHAM.

We next assayed killing of strains J1, J5 and J9 by freshly harvested human neutrophils. SCV strains J5 and J9 were more resistant than J1 to neutrophil killing (12.6 ± 2.7% and 9.6 ± 2.6%, respectively, *vs*. 31.8 ± 1.8; one-way ANOVA with Dunnett's multiple comparison tests (J1 versus J9, 95% confidence interval of difference [11.25-27.2], $p < 0.0001$, and J1 versus J9 [14.2 to 30.2], $p < 0.0001$) (Supplementary Data 2). J9 was also more resistant to paraquat (survival of 77.3 ± 5.2% at 100 mM and 19.9 ± 5.6% at 165 mM), a redox-active drug that generates endogenous superoxide anions, than J1 (survival of 33.1 ± 5.8% at 100 mM and 3.8 ± 0.9% at 165 mM; $p = 0.0005$ ($t = 5.7$, df=7.9) and $p = 0.04$ ($t = 2.9$, df = 4.2), respectively; student's t test).

Studies have shown that dysfunctional mitochondria activate *PDR1* and, in turn, *CDR1*[21]. As expected from these reports, *PDR1* and *CDR1* were up-regulated by a mean ± standard error of 14.6 ± 2.0-fold and 257.1 ± 5.9-fold, respectively, in strain J9 compared with strain J1 (RT-PCR). Along these lines, fluconazole and voriconazole MICs were higher against strains J6-J10 (>64 μg/mL and >16 μg/mL, respectively) than they were against J1-J4 (4−8 μg/mL and 0.5 μg/mL, respectively)

(Supplementary Table S1). Fluconazole and voriconazole MICs against J5 (32 μg/mL and 2 μg/mL, respectively) were intermediate to those against the other strains.

Finally, we compared strains J1 and J9 for virulence during hematogenously disseminated infections of mice. None of the mice died at day 21 following lateral tail vein injections of $1.5 × 10^8$ CFU of either strain. For tissue burden studies, mice were infected with $1 × 10^7$ CFU. J9 caused significantly lower tissues burdens than J1 in kidneys and spleens at 1, 3, and 7 days, and in livers at 3 and 7 days (Fig. 7).

We then focused on *C. glabrata* relapsing strain J11. Strain J11 was a SCV that demonstrated slow growth in YPD media, inability to grow in YP glycerol, small size on SDA plates, purple colonies on eosin plates, and resistance to fluconazole, voriconazole and posaconazole. J11 clustered with other J strains by nuclear genome phylogeny (Fig. 8a). By mitochondrial phylogeny, J11 clustered with SCV strains, and was closest to strain J8 (Fig. 8b).

To assess if *C. glabrata* genotypic and phenotypic diversity might arise during growth in vitro, we spiked a sterile blood culture bottle with index strain J1. After 3 days of incubation, turbidity was observed. Aliquots sub-cultured on SDA plates revealed homogenous colonies

| Azole MIC (µg/ml) | L1 | L2 | L3 | L4 | L5 | L6 | L7 | L8 | L9 | L10 |
|---|---|---|---|---|---|---|---|---|---|---|
| FCZ | 16 | 32 | 32 | >256 | 32 | 64 | 16 | 256 | 128 | 64 |
| VOR | 0.25 | 0.25 | 0.5 | 8 | 0.5 | 0.5 | 0.25 | 4 | 0.5 | 0.5 |
| POS | 0.125 | 0.125 | 0.125 | 4 | 0.125 | 0.125 | 0.125 | 4 | 0.125 | 0.125 |
| ISAV | 0.125 | 0.125 | 0.125 | 4 | 0.125 | 0.125 | 0.125 | 4 | 0.125 | 0.25 |

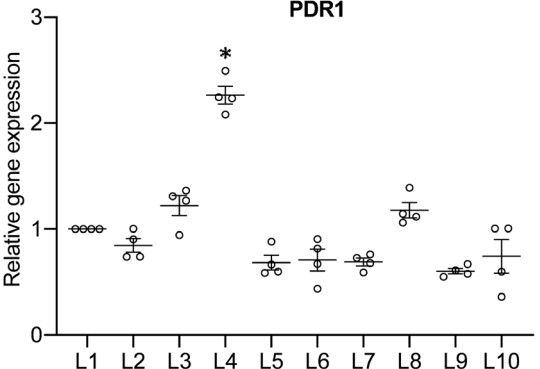
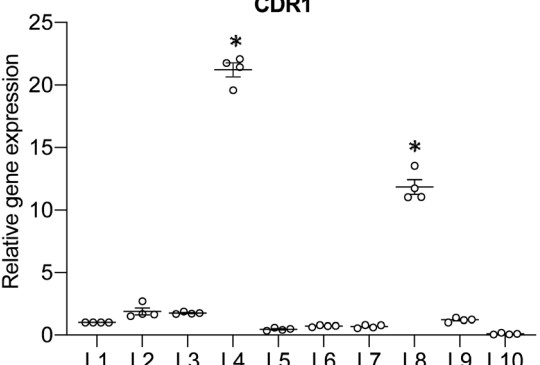

**Fig. 4 | Azole susceptibility and *PDR1* and *CDR1* expression data for *C. glabrata* strains from patient L.** Strains L4 and L8 were more resistant to all 4 azole agents tested than were the other 8 L strains. Y-axis represents fold difference of *PDR1* or *CDR1* expression by respective L strains relative to the index strain L1, normalized to 18 S rRNA. Data on graph are presented as mean ± standard error of mean (SEM) of relative gene expression of *PDR1* or *CDR1* from quadruple experiments. Statistical analyses were performed using one-way ANOVA with Dunnett's multiple comparisons test comparing the mean rank of relative gene expression of L1 to other L strains. By RT-PCR, relative expression of *PDR1* by strain L4 was significantly higher than that by strain L1 (mean rank difference of 1.3 (95% confidence interval (CI) of difference: 0.9 to 1.6), *p* < 0.0001). Relative expressions of *CDR1* by strains L4 and L8 were significantly higher than that by strain L1 (L4 vs L1: mean rank difference of 20.2, 95% CI: 19.1 to 21.3, *p* < 0.0001; L8 vs L1: mean rank difference of 10.8, 95% CI: 9.7 to 11.9, *p* < 0.0001). * denotes that the difference is statistically significant (*p* < 0.0001).

consistent with NCVs. There was no evidence of SCVs after 5 days of incubation. Ten strains from randomly selected colonies had growth rates and susceptibility to fluconazole, voriconazole and posaconazole that were similar to those of strain J1. Strains obtained from spiked blood culture demonstrated at least 2-fold fewer nucleotide differences than strains J1-J10 (for both nuclear and mitochondrial genomes). We analyzed variants associated with non-synonymous SNPs and indels that discriminated within-patient strains J1-J10 and those that discriminated strains recovered from the spiked blood culture bottle. There were 115 non-synonymous variant-containing genes that discriminated among J1-10 that were not identified in the spiked culture bottle (Supplementary Table S4). GO term analysis of these variants revealed genes enriched in processes linked to mitochondrial functions involved in respiratory electron transfer and ATP synthesis (Supplementary Table S4). There were 69 non-synonymous variant-containing genes that discriminated among both J1-J10 and strains from the spiked bottle; genes encoding cell wall proteins involved in adhesion were over-represented. Only 20 non-synonymous variant-containing genes were present uniquely in strains from the spiked sample; no biologic process or function was significantly over-represented among these genes.

## Discussion

In this study, we demonstrated that genotypic and phenotypic diversity of a *Candida* species was common in blood cultures of individual patients with BSI. We showed that positive blood cultures from each of 10 patients diagnosed with *C. glabrata* BSIs harbored clonal but genetically diverse strains that differed by SNPs and chromosomal rearrangements, and, to a lesser extent, insertions, deletions, and presence or absence of specific genes. In 2 patients, genetically distinct strains exhibited unique phenotypes that were potentially important under antifungal selection pressure and during BSIs, including differences in susceptibility to antifungal agents and phagocytosis. In both patients, blood cultures were comprised of mixed fluconazole-susceptible and -resistant populations, but the clinical microbiology laboratory only identified a fluconazole-susceptible index strain. Diversity in drug susceptibility was likely clinically relevant, as fluconazole-resistant *C. glabrata* strains were subsequently recovered by the clinical laboratory during persistent BSI (patient L) or relapsing invasive candidiasis (patient J). In one patient (J), fluconazole-resistant strains were respiration-deficient SCVs, which carried mitochondrial genome mutations that were not present in fluconazole-susceptible, NCV strains. SCVs were also not identified by the clinical laboratory during the initial BSI. Taken together, our data challenge the long-standing, "single organism" model of pathogenesis, and suggest a population-based model of *C. glabrata* genotypic and phenotypic diversity during BSIs.

Most studies of *Candida* diversity at a site of infection have characterized longitudinal, rather than contemporaneous strains. WGSs of *C. glabrata* or *C. albicans* strains from serial oral, vaginal, blood, stool and respiratory cultures revealed within-patient differences in SNPs, indels, gene CNVs, aneuploidies, and loss of heterozygosity (LOH, for *C. albicans*)[5,7,22–26]. Longitudinal emergence of antifungal resistance, often associated with appearance of resistance-conferring mutations, is well-recognized among *C. glabrata*[26,27]. In contrast, there are few studies of *Candida* diversity from a single site at a given timepoint. In a *Candida auris* outbreak investigation, probabilistic analysis of WGS data from 6 to 12 pooled colonies suggested mixed colonization or disease in ~25% of patients;[28] it is unclear if heterogenous populations were identified from blood cultures. In another study, WGSs of strains from 3 independent *C. albicans* colonies from oral cultures of healthy volunteers clearly demonstrated that each strain was unique, mostly due to SNPs and short-range LOH[29]. In a multi-center study, 5.6% and 8.1% of positive *Candida* cultures from blood and any clinical site, respectively, had antifungal polyresistance, defined as heterogenous susceptibility testing results among 5 independent colonies[30]. Such mixed populations were detected in 15.3% of *C. glabrata*-positive clinical samples, a frequency in keeping with our description of unrecognized azole-resistant strains in 2 of 10 patients.

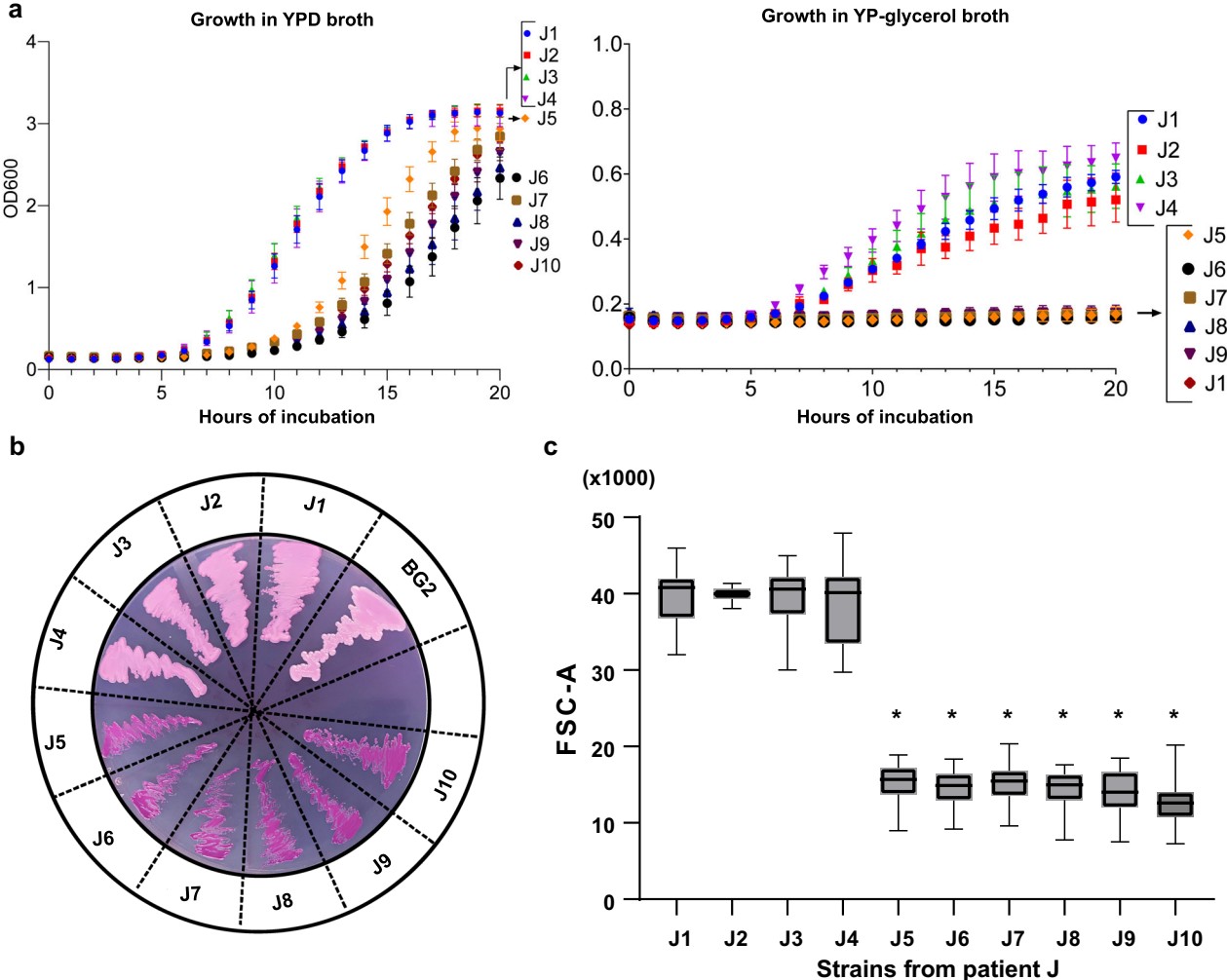

**Fig. 5 | Phenotypes of normal colony and small colony variants in J strains.**
**a** Growth rates of J strains in Yeast Extract-Peptone-Dextrose (YPD, left panel) and Yeast Extract-Peptone-glycerol (YP-glycerol, right panel) broth. Cells were incubated in respective media at 30° C. Growth of strain J5 in YPD was intermediate to that of strains J1-J4 (normal colony variants) and J6-J10 (small colony variants). J5 grouped with J6-J10 by not growing in YP-glycerol. Note the difference in Y-axis scales for the graphs. Data presented are mean ± standard error of mean from triplicate experiments. **b** Morphology of J strains on eosin Y plates. Normal colony variant strains were light pink, whereas small colony variant strains were dark pink to purple. Strains J1-J4 and J5-J10 stained as normal and small colony variants, respectively. Control strain *C. glabrata* BG2 stained as a normal colony variant. **c** Cell sizes of J strains, as measured by FACS. Cells in stationary phase were washed prior to measurement. Forward versus scatter (FSC and SSC) density plot was performed, and the cell population of interest was identified with polygon gating that encompassed at least 95% of all cell population. Box and whisker plots of FSC-A are used to represent cell sizes. The upper and lower whiskers represent maximum and minimum values, respectively; the hinges of the box represent 25% to 75% percentiles and the horizontal line within the box represents median values. The data shown are combined FSC-A data from 3 independent experiments. The mean rank of FSC-A of the index strain J1 was compared with those of the other J strains, and *p*-values determined using Kruskal-Wallis test with Dunn's multiple comparisons. The mean rank of J1 FSC-A was significantly higher than those of J5 (mean rank difference of 54,773), J6 (60,758), J7 (57,307), J8 (61,710), J9 (64,492) and J10 (76,495); *p*-values < 0.0001.

Contemporaneous strains in the multicenter study did not differ by multilocus sequence type, but WGS was not performed[30].

*C. glabrata* and most pathogenic *Candida* spp. are GI tract commensals. Genotypically and phenotypically diverse bacterial populations are increasingly recognized during colonization and chronic infections of non-sterile sites, including GI tract, lungs, and skin[13–15]. Aside from our recent demonstration that positive blood cultures of patients with carbapenem-resistant *K. pneumoniae* BSIs were comprised of genetically variant, clonal strains that differed in antibiotic susceptibility and virulence[16], there are scant data on microbial diversity during acute infections of putatively sterile sites[31]. Numbers of nucleotide differences (i.e., SNPs, indels) between contemporaneous *C. glabrata* BSI strains here (pairwise average per-patient: ~3500) were broadly comparable to those previously reported among contemporaneous *C. albicans* from oral cultures (~500-5,100 SNPs)[29]. Mutations in our strains were predominantly synonymous and found in non-coding regions. Similar findings in a previous study of *C. albicans* passed in vitro and in vivo were felt to reflect purifying selection that limited accumulation of mutations in protein-coding sequences[3]. More recently, however, investigators were surprised to find enrichment of non-synonymous, coding sequence mutations among serial *C. glabrata* BSI strains[26]. Reasons for discrepancies between studies are unclear, and merit further exploration.

In keeping with previous data for *C. glabrata*, we found that genes encoding adhesins and other cell wall proteins such as those in EPA, AWP and PWP families were over-represented as sites of non-synonymous mutations in strains from patients' blood cultures[3,7,8,32]. Genes encoding adhesins and cell wall proteins were over-represented among non-synonymous variants recovered from both patient J's blood culture and a culture bottle spiked in vitro with strain J1. Within-patient SNPs were disproportionately concentrated within repeat-rich, subtelomeric regions, which are well-recognized

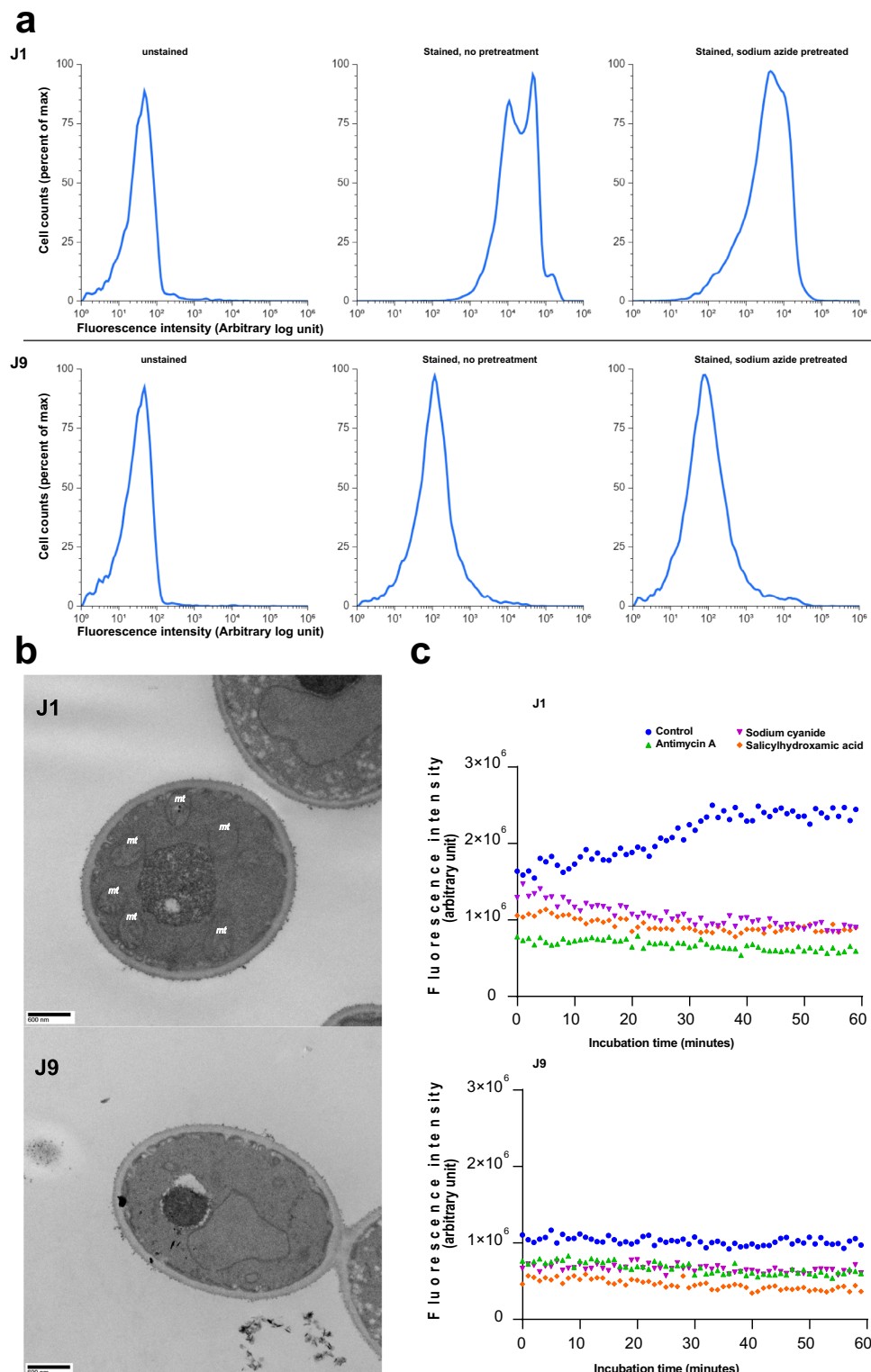

mutational hotspots that are home to prominent adhesin gene families. Previous studies of longitudinal *C. glabrata* BSI strains have reported mutations in adhesins and cell surface proteins[26]. Although expansions of adhesin gene families are common in *C. glabrata*[33,34], CNVs were infrequent among contemporaneous strains here. In contrast, we observed chromosomal rearrangements among strains from 8 of 10 patients. Chromosomal rearrangements are frequent and highly dynamic in *C. glabrata*[8,35], and they are postulated to represent adaptive mechanisms for survival in the host[36]. Large-scale,

structural genome variations like aneuploidies or polyploidies were not detected among strains from our patients.

Azole resistance in various *C. glabrata* strains from patients L and J was associated with increased expression of transcriptional regulator gene *PDR1* and, in turn, up-regulation of efflux gene *CDR1*. Azole-resistant strains from patient L harbored a previously uncharacterized G346C mutation in the Pdr1 central regulatory domain[17]. Using site-directed mutagenesis, we demonstrated that G346C is a gain-of-function mutation that results in *PDR1* hyperactivation and enhanced

**Fig. 6 | Phenotype comparisons of normal colony variant (NCV) J1 and small colony variant (SCV) J9. a** Flow cytometry of strains J1 (upper panel) and J9 (lower panel) stained with rhodamine 123. Cells ($1 \times 10^6$ CFU/mL) were grown in 4 mL of YPD broth overnight at 30 °C. Thereafter, cells were stained with rhodamine 123, with or without 1 mM sodium azide pre-treatment. Rhodamine-123 fluorescence intensity is presented on the X-axis; intensity increases from left to right. The Y-axis presents cell numbers, as the percentage of the total number of cells. FSC and SSC density plot was performed, and the cell population of interest was identified with polygon gating. Histogram bins are normalized to peak. Unstained J1 and J9 cells (a and b, respectively) are controls. J1 and J9 cells stained with rhodamine 123 in absence of sodium azide pre-treatment are presented in c and d, respectively. In absence of sodium azide treatment, J1 (c) exhibited greater fluorescence than did J9 cells (d), consistent with higher mitochondrial activity. J1 and J9 cells stained with rhodamine 123 following sodium azide pre-treatment are presented in e and f, respectively. Set of figures below are representative images from one of two

independent experiments that showed similar findings. **b** Transmission electron micrographs of strains J1 and J9. Representative images (from three independent experiments, with 30 images taken per experiment) shown here highlight reduced numbers and aberrant morphology of mitochondria in small colony variant strain J9, compared to normal colony variant J1. Several normal appearing mitochondria are denoted by *mt*. **c** Evaluation of respiratory status of strains J1 and J9 using electron transport inhibitors. Cells grown in synthetic completed medium for 48 h at 30° C were untreated or exposed to sodium cyanide 100 mM, salicylhydroxamic acid (SHAM) 100 mM or antimycin 1 μM. Oxygen consumption was measured as phosphorescent probe signal intensities (y-axis) *versus* time (x-axis). In absence of drug exposure (control), oxygen consumption by strain J1 was greater than that by strain J9. Each of the electron transport inhibitors resulted in significant reduction in J1 oxygen consumption, without impacting consumption by respiratory-deficient strain J9. Figures below are representative images from one of two independent experiments that showed similar trend of response to inhibitors.

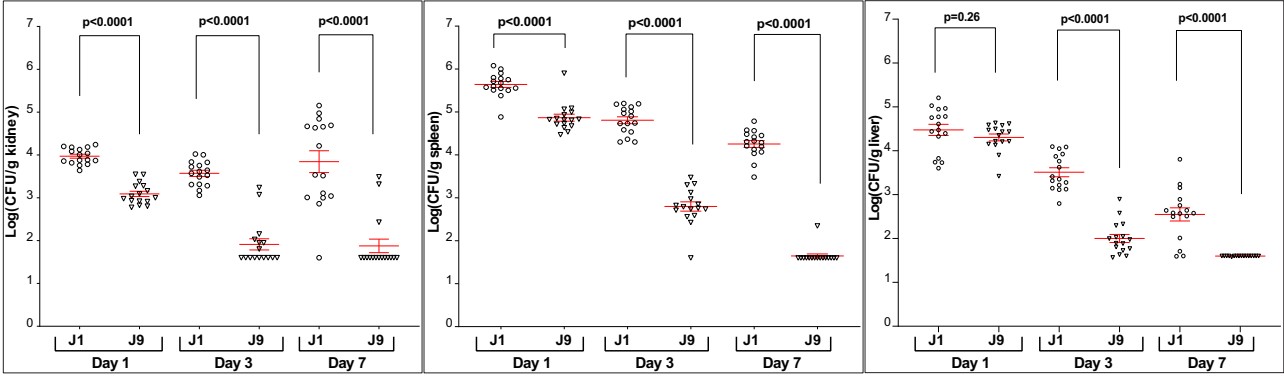

**Fig. 7 | Tissue burdens of *C. glabrata* strains J1 and J9 in kidneys, spleens and livers of mice during hematogenously disseminated infections.** Sixteen mice per group (8 male, 8 female) were infected intravenously (IV) with $1 \times 10^7$ CFU of respective strains (J1, empty circle; J9 inverse triangle), and they were sacrificed for CFU enumeration at days 1, 3 and 7. $\text{Log}_{10}$ tissue burdens per organ are presented in scatter dot plots below; red lines represent mean of log (CFU/gram of tissue) ± standard error of mean. Strain J9 was significantly attenuated for virulence in all target organs. P-values (two-tailed) for pair-wise comparison were determined

using student's t tests with Welch's correction. For kidney tissue burdens, Welch-corrected t and degree of freedom were 11.34 and 27.63 at day 1, respectively; 11.14 and 23.25 at day 3, and 6.56 and 25.02 at day 7, respectively. Corresponding data for spleens were 7.25 and 29.24 at day 1, 14.92 and 27.55 at day 3 and 28.06 and 24.08 at day 7; corresponding liver data were 1.15 and 25.0 at day 1, 11.01 and 29.65 at day 3, and 6.32 and 15 at day 7. Overall, there were no significant differences in tissue burdens between male and female mice.

downstream *CDR1* activation. Azole-resistant strains from patient J carried wild-type *PDR1*, but they exhibited SCV phenotypes including slow growth on enriched medium, attenuated proliferation in non-fermentable carbon sources, and mitochondrial and aerobic respiratory defects. Mitochondrial derangements have been shown to activate the pleiotropic drug resistance Pdr pathway that globally upregulates expression of drug efflux pump genes (e.g., *CDR1*)[17,21,37–39]. The most striking azole resistance and SCV phenotypes were in strains with the most SNPs in mitochondrial genes, regardless of whether they had normal (J6) or reduced (J7-J10) numbers of mitochondria. NCV strains J1-J4 had normal numbers of mitochondria and few mitochondrial gene mutations. SCVs were likely missed by the clinical laboratory because they were not visualized on blood or SDA agar plates until ≥84 h after sub-culture. Therefore, SCV prevalence may be underestimated unless culture plates are assiduously monitored for several days after *Candida* growth is detected. Our finding has clinical significance since patient J subsequently had a relapsing infection due to a fluconazole-resistant SCV strain (J11) that clustered phylogenetically with SCVs from the initial blood culture.

Mitochondria possess their own genetic material that evolves independently from the nuclear genome. In certain *C. glabrata* strains, the mitochondrial genome is hyper-diverse compared to the nuclear genome[26]. On the whole, we found that mitochondrial genome phylogeny was less sensitive than WGS phylogeny in identifying within-patient strain diversity. Nevertheless, data for J strains indicate that mitochondrial genome analysis may be a useful complement to WGS

phylogeny for *C. glabrata* with apparent growth or respiration deficiencies, SCV phenotypes, or decreased antifungal susceptibility. There was significant over-representation of non-synonymous variant-containing genes involved in mitochondrial functions among strains J1-J10, including 115 genes identified in patient J's blood culture but not in a spiked culture bottle. Mitochondrial phylogeny accurately distinguished NCV from SCV J strains, whereas as nuclear phylogeny did not. It is unclear what precipitated mitochondrial dysfunction and emergence of SCVs. Azole exposure has been linked to mitochondrial damage[40–44], and patient J received 28 days of fluconazole prior to BSI. The relatively few clinical *C. glabrata* SCV strains reported to date were recovered from azole-experienced patients[41,45]. Alternatively, mitochondrial damage could have stemmed from oxidative stress from host phagocytes[38]. Regardless of mechanism, dysfunctional mitochondria activate compensatory responses that confer adaptive advantages with cross-resistance to both azole and phagocytic killing[21,37,38]. Compared to NCV strain J1, SCV strain J9 was relatively resistant to neutrophil killing, but it was significantly attenuated for virulence in mice with hematogenously disseminated candidiasis. SCV strains had growth defects under various conditions in absence of azole exposure, which likely offset potential advantages afforded by evasion of phagocytosis.

Previous studies of virulence of *C. glabrata* SCVs have yielded conflicting results. In one report, an SCV generated by ethidium bromide treatment exhibited reduced virulence during murine disseminated candidiasis[46]. In another study, however, a fluconazole-

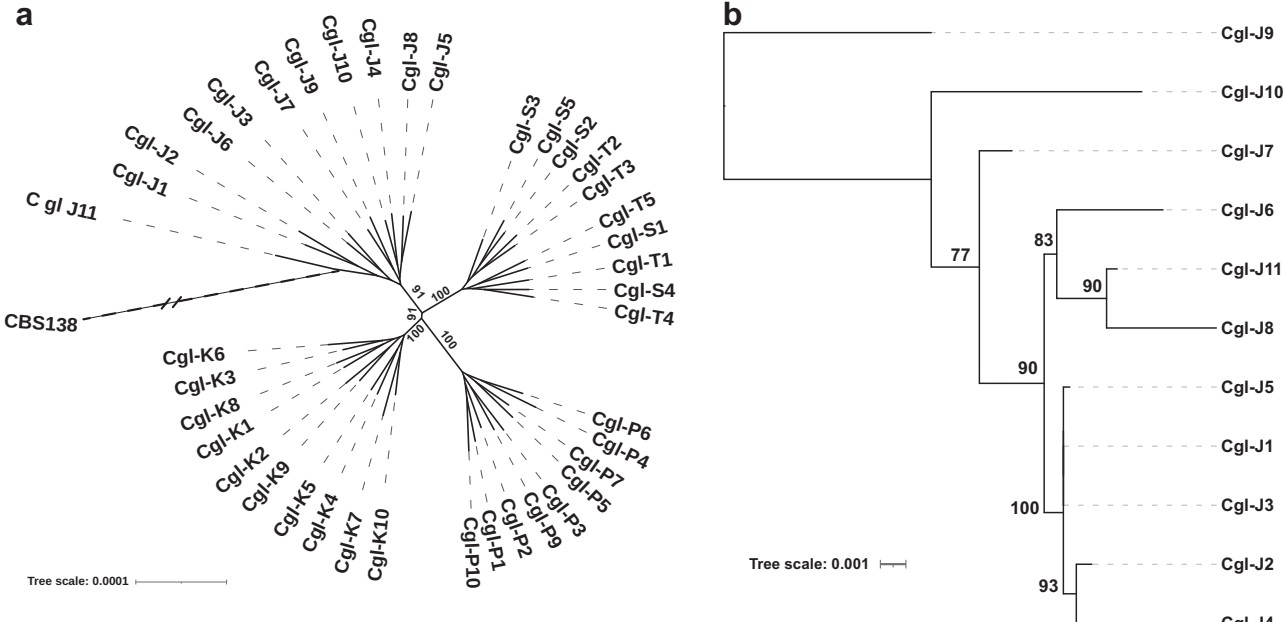

**Fig. 8 | Phylogenomic analysis of ST3 *C. glabrata* strains, including relapsing strain J11. a** Radial unrooted whole genome phylogeny of strains from patients J, K, P and ST (i.e., ST3 strains). Strains cluster by patient. Note that strain J11 clustered with J1-J10. *C. glabrata* CBS138 is included for reference. **b** Mitochondrial genome phylogeny of 11J strains. Note that NCV strains (J1-J5, on bottom) and SCV strains (J6-J10 and J11, on top) cluster with each other. J11 is closest to J8 by mitochondrial genome phylogeny.

resistant oropharyngeal *C. glabrata* SCV was more virulent than an antecedent fluconazole-susceptible strain[41]. We previously reported that a *C. albicans* SCV that emerged after hematogenous passage through mouse organs caused lower mortality and acute tissue burdens during disseminated candidiasis than its parent strain, but it persisted within tissue for a prolonged period[20]. Results are difficult to compare between studies since strains were not isogenic and they were created through different methods. Virulence is a complex and multi-factorial phenomenon, which can vary based on site of infection and conditions within a given host. Moreover, *C. glabrata* is a relatively weak pathogen in mice, and limited virulence in mouse infection models does not necessarily correspond to lack of pathogenicity in humans. For example, SCV strain J9 was less virulent than NCV J1 following IV inoculation of mice, but SCVs nevertheless persisted in patient J after resolution of candidemia and re-emerged to cause relapsing infection. The concept of virulence is particularly complex for an opportunistic pathogen like *C. glabrata* that has significant redundancy in virulence determinants, lacks a dominant virulence factor, and often causes diseases in patients with immunodeficiencies and other host defense impairments.

Our study was designed to collect *C. glabrata* as blood cultures were being processed according to standard clinical microbiology lab practices. As such, a strength of the study is that results reflected *C. glabrata* diversity that might be unrecognized by clinicians. We acknowledge that we cannot definitively determine when specific mutations may have arisen. However, our data suggest that most within-patient *C. glabrata* diversity emerged in vivo, rather than in vitro. Indeed, there were twice as many nucleotide differences among strains from patient J's blood culture than there were among strains from a culture bottle spiked with strain J1 (p < 0.0001). Furthermore, SCV strains did not emerge in the spiked culture. It is unclear whether BSI diversity may have stemmed from one-time inoculation of the blood with a mixed population of strains from sites of colonization such as the GI tract, serial inoculation of strains, and/or mutations emerging in the bloodstream prior to collection of cultures. We believe the majority of diversity that arose in the host was likely to do so at

colonization sites since *Candida* population sizes are typically greater and durations of infection are longer during colonization than they are for candidemia. Furthermore, in WGS analyses of *C. glabrata* from longitudinal cultures of various body sites over 0-90 days, investigators concluded that inter-strain genetic variation was primarily due to standing, pre-existing diversity within the population rather than to accumulation of new mutations[7].

There are some limitations to our data. We sequenced only 10 strains per blood culture bottle, which might not adequately represent the entire *C. glabrata* population. In the future, metagenomic sequencing performed directly on DNA recovered from the bloodstream might afford more comprehensive coverage of microbial variants. At present, this approach is limited by relatively low concentrations of microbial DNA, and challenges in assigning sequence variations to individual strains[16]. We only sequenced strains from the first episode of BSI. Future studies of longitudinal samples may provide insights into strain evolution, population changes and importance of pre-existing *vs.* de novo antifungal resistance. Our results cannot be extrapolated to BSIs by other *Candida* spp. or other pathogens. The extent of diversity we detected may reflect particular features of our patients, who were severely ill and who had extensive past medical histories. Six patients had polymicrobial BSIs that included pathogens in addition to *C. glabrata*. Moreover, 5 patients had recent antifungal exposure or breakthrough BSIs while receiving an antifungal agent, and 2 patients had invasive candidiasis within the previous 90 days. Data here and from our earlier investigation of carbapenem-resistant *K. pneumoniae* suggest that genotypic and phenotypic microbial diversity may be especially relevant to BSIs by enteric opportunistic pathogens.

In conclusion, we identified genotypic and phenotypic diversity among *C. glabrata* from blood cultures of individual patients. Our findings suggest that *C. glabrata* diversity arising in response to selective pressures during commensalism, including that imposed by antifungal therapy as in several patients here, may result in variant strains that are better able to cause opportunistic infections. Building upon our data for *C. glabrata* and *K. pneumoniae*, follow-up studies are

warranted to investigate BSIs by other GI enteric opportunistic pathogens, including different *Candida* spp., and to compare diversity within the GI tract with that encountered during BSIs. A pressing question is whether microbial diversity has broad clinical significance, as implied by the unrecognized antifungal resistance in patients L and J. Specific issues for further study, in addition to those raised above, include defining how often clinical laboratories fail to identify antifungal resistant variants (i.e., heteroresistance), whether such events lead to treatment failures, and the relative importance of pre-existing *versus* de novo resistance. Results of such studies will determine whether clinical laboratory practices and treatment decision-making should consider microbial populations at sites of infection. The European Committee on Antimicrobial Susceptibility Testing currently recommends measuring antifungal MICs against suspensions of up to 5 representative colonies (http://www.eucast.org/ast_of_fungi), but this practice has not been endorsed by the Clinical and Laboratory Standards Institute[47]. In the long term, basic research directed toward understanding how microbial diversity and adaptation facilitate commensalism and pathogenesis of different types of invasive infections may identify strategies for preventing and ameliorating disease.

## Methods

### Ethics statement
This study complies with relevant ethical regulations. It was approved by the University of Pittsburgh Institutional Review Boards through an "Expedited Review procedure" as published by the OHRP, 45 CFR 46.110 and FDA 21 CFR 56.110 (IRB protocol #22040147). The need for informed consent was waived for this project deemed to pose minimal risk to patients. We utilized remnant blood cultures and clinical isolates from the UPMC clinical microbiology laboratory at the time they were discarded (the blood cultures were drawn purely for clinical care purposes). Clinical characteristics were identified through limited retrospective review of electronic medical records. The mouse experiments were approved by the University of Pittsburgh Animal Care and Use Committee (protocol # IS00018518). Mice were provided with water and a standard laboratory diet *ad libitum*. They were supplied with hardwood chips as bedding and housed in a temperature-controlled, air-conditioned room on a 12-hour light-dark cycle.

### Clinical strains and growth conditions
Blood culture bottles from patients with *C. glabrata* BSI were obtained from the UPMC clinical microbiology laboratory, as were the strains used for species identification and susceptibility testing ("index strains"). We streaked 10 μL from the positive blood culture bottle from each patient onto 2 Sabouraud dextrose agar (SDA, Fisher Scientific, DF0109-07-07-3) plates and incubated overnight at 35 °C. *C. glabrata* were confirmed by matrix assisted laser desorption ionization-time of flight mass spectrometry. A strain isolated from each of 4–9 morphologically indistinguishable colonies from each patient underwent Illumina NextSeq whole genome sequencing (WGSing). Index strains underwent both short-read Illumina WGS and Oxford Nanopore (ONT) using MinION sequencing.

### DNA extraction and sequencing
For short-read sequencing, cells were grown in Yeast Extract-Peptone-Dextrose (YPD; Yeast Extract: BD Difco 210929, Peptone: BD Difco 11677, Dextrose: Fisher Scientific D16-500) at 30 °C overnight, then harvested and resuspended in 1.2 M sorbitol (Fisher Scientific BP439-500), 50 mM EDTA (Sigma E8008). DTT (Sigma 11583786001) and lyticase (Fisher Scientific NC1644419) were added to the suspension to 10 mM and 85 U/reaction, respectively. Cells were incubated at 37 °C for 45 min to make spheroplasts, which were harvested for genomic DNA extraction using Qiagen's DNeasy Blood and Tissue Kit (Qiagen 69504, Hilden, Germany). For long-read sequencing, high molecular weight genomic DNA from overnight grown cells was extracted following the protocol of Denis et al[48]. DNA sequencing libraries were prepared with the Nextera XT (Illumina, San Diego, USA) protocol according to the manufacturer's instructions. Oxford Nanopore libraries were prepared using the Rapid (Oxford Nanopore, Oxford, UK) protocol per manufacturer's instructions. Illumina libraries were pooled with unique barcodes and sequenced to at least 50X coverage with 2×150 bp paired end reads. Oxford libraries were pooled with unique barcodes and sequenced to at least 30X coverage.

### Bioinformatic analyses
Real time analysis (RTA) v3.4.4 software was used for base calling. Raw read zipped files were analyzed through a pipeline built using NextFlow[48], which integrates Burrows-Wheeler Aligner (BWA)[49] for reads mapping to the *C. glabrata* CBS138 genome sequence[50,51], downloaded (on 4/30/2020) from *Candida* Genome Database[51]. The Genome Analysis Toolkit (GATK)[52] was used for deduplicating and sorting of mapped reads, variant calling and filtering after a base quality score recalibration (BQSR). SnpEff was employed for variant annotation[53]. VCF variant files from all strains were merged using BCFTOOLS[54], and a whole genome multiple sequence alignment (MSA) was extracted from the multi-sample VCF file using VCF2MSA python script (https://github.com/tkchafin/vcf2msa.py). All aligned contigs were then merged and converted to a phylip format using a perl script (https://github.com/nylander/catfasta2phyml). A phylogenomic tree was built using FastTree[55] or RaxML[56]. The tree topology was manipulated in Geneious (Biomatters Ltd, Auckland, New Zealand), iTol v6[57], or Adobe Illustrator (Adobe Inc., San Jose, CA). We determined the sequence type (ST) for all strains using PubMLST (https://pubmlst.org/). We selected gene variants that were called in one or more strains, but not all strains within each patient. We focused on variants associated with non-synonymous amino acid substitutions, disrupted start or stop codons, and frameshift insertions or deletions. Genes corresponding to variants were analyzed for enrichment in the *Candida* Genome Database[51].

Gene deletions/duplications were analyzed using two strategies. The first employed GENOME-Strip[58] and was based on coverage depth in addition to information from split reads, paired reads, and the assembly of reads. In the second strategy, we employed a Nextflow pipeline that performs de novo assembly of long reads from the representative of each patient's strains using FLYE[59]. Short reads were mapped to the representative strain's assembly using BWA, and Pilon[60] was used for error correction and to polish and produce the final assembly. Assembled genomes within each patient were aligned using progressiveMauve[61]. Alignments were visually inspected in Mauve to detect gene deletions/duplications. D-Genies was used to generate genome similarity dot plots[62].

To estimate mitochondrial copy number for each strain, we first used GSUTILS scripts from GENOME-STRIP. We then used quantitative PCR, with *COX1* as target for a mitochondrial gene and *ACT1* for control (Supplementary Table 5). Amplification was performed using Maxima SYBR Green qPCR Master Mix (Fisher Scientific FERK0223).

All programs in this project were run using University of Pittsburgh Center for Research Computing resources.

### Reverse transcription-quantitative PCR
RNA was harvested from yeast cells in exponential growth in YPD broth using the RiboPure RNA purification kit for yeast (Invitrogen AM1924), and then treated with DNaseI. cDNA was synthesized using Verso cDNA synthesis kit (Fisher Scientific AB1453A). Genomic DNA contamination was checked by PCR with primers flanking the intron of *C. glabrata ACT1*. Primers used throughout the study are listed in Supplementary Table S5[46,63,64]. Quantitative PCR with SYBR Green qPCR Master Mix (Fisher Scientific FERK0223) was performed with 1:10 diluted cDNA. Target gene expression was calculated using the ΔΔCT method, with

normalization to housekeeping genes *ACT1* and Cg18S. Data are shown as mean with standard error of mean (SEM) of at least 3 independent experiments.

## Construction of *PDR1* mutants

To introduce a G346C mutation, *PDR1* was first disrupted using the SAT1 flipper method (Supplementary Table S5)[65]. ~500-bp proximal (F1) and distal (F2) gene fragments of the target region were cloned into pSFS2A (plasmid kindly provided by Morschhäuser)[65], which was linearized by digestion with *Kpn*I and *Sac*I (New England Biolabs R3142S and R3156S, respectively). Deletion cassette (F1-SAT1-F2) was introduced into *C. glabrata* BG2 and L4 cells by electroporation. Correct integration was verified by PCR. SAT1 cassette was recycled by FLP-mediated excision. For *PDR1* replacement, the complete *PDR1* ORF flanked by 500 bp was amplified by PCR from genomic DNA of BG2 (with wild-type *PDR1*) and L4 (with G346C *PDR1*). PCR products were ligated to pSFS2A-F2. The cloned *PDR1* sequence was confirmed by Sanger sequencing. Resulting plasmids were linearized by *Kpn*I and *Sac*I and transformed into strains BG2 Δ*PDR1* and L4 Δ*PDR1*.

## Transmission Electron Microscopy

Microscopy was performed at the University of Pittsburgh Center for Biologic Imaging (CBI). Cells were fixed in cold 2.5% glutaraldehyde (Polysciences BLI1909) in 0.01 M phosphate-buffered saline (PBS), pH 7.3 (Fisher Scientific R582350010F), post-fixed in 4% aqueous potassium permanganate (Sigma 7722-64-7), then stained with 2% aqueous uranyl acetate (Electron Microscopy Sciences 6381-92-6). Cells were dehydrated through a graded series of ethanol (Fisher Scientific BP82014 and BP2818100), and propylene oxide (Electron Microscopy Sciences 75-56-9), and embedded in Poly/Bed® 812 (Polysciences 08792-1). Ultrathin sections (65 nm) were stained with 2% aqueous uranyl and Reynold's lead citrate (Lead Nitrate, Sodium Citrate and Sodium Hydroxide, Fisher Scientific 50-268-90), and examined on JEOL 1400 Plus transmission electron microscope (JEOL Peabody, MA) with AMT Capture Engine Software (version 602.600.652).

## Flow cytometry

Respiratory status of cells was investigated by flow cytometry[66]. Yeast cells in stationary phase ($2 \times 10^6$/mL) were incubated with rhodamine 123 (Invitrogen R302) at a final concentration of 10 µg/mL for 30 min at 37° C. To inhibit electron flow of the respiratory chain, cells were pre-incubated with 1 mM sodium azide (Fisher Scientific 26628-22-8) for 2 h before the addition of rhodamine 123. Cell fluorescence was quantified with a FACScan flow cytometer. For cell size measurement, unstained yeast cells were washed and evaluated using FACScan. Forward and side scatter gating (FSC-A and SSC-A, respectively) were recorded for 10,000 events. Data were collected on an LSRFortessa with BD FACSDiva (v8.0.2), analyzed using FlowJo v10.9 software, and visualized using the online tool Floreada.io (https://floreada.io/analysis). Gating analysis of flow cytometry data was performed using SpectroFlo® (version 3.0.0).

## Phenotype assays

**Respiration.** Respirometry was carried out using the Oxygen Consumption Rate Assay Kit (Caymanchem 600800). *C.glabrata* strains were grown in synthetic complete medium (0.67% of Bacto-Yeast nitrogen base w/o amino acids (Fisher Scientific BD 291920), 2% glucose, 0.2% amino acids drop-out (Fisher Scientific BD 630415)) for 48 h at 30° C. After wash with PBS buffer, 140 µl of yeast ($5 \times 10^8$/mL in 1X PBS with 2% glucose) were added to wells of 96-well Costar flat bottom plate (Corning 3596) containing inhibitors sodium cyanide 100 mM, salicylhydroxamic acid (SHAM, Fisher Scientific AC132620250) 100 mM, or antimycin (Caymanchem 600800) 1 µM. After overlay with 100 µl oil, plates were measured kinetically with SpectraMax i3x (Software Pro 6.4) at excitation 380 nm, emission

650 nm for 90 minutes. Three independent experiments were performed. Phosphorescent oxygen probe signal intensities were plotted *versus* time.

**Antifungal susceptibility testing.** Antifungal susceptibility testing was performed using the broth dilution technique according to the Clinical and Laboratory Standards Institute (CLSI) standardized method[47]. Concentrations ranged from 0.25-256 µg/mL for fluconazole (Fisher Scientific 86386-73-4), and 0.015-16 µg/mL for voriconazole and posaconozole (Fishser Scientific (TCI America products) VO116 and P2477, respectively), and isavuconazole (BOC Sciences 241479-67-4). For fluconazole, strains were classified as susceptible-dose-dependent or resistant (MIC ≤ 32 µg/mL and ≥64 µg/mL, respectively) according to CLSI criteria[47]; for purposes of this manuscript, we refer to susceptible-dose-dependent as susceptible. Currently there are no defined voriconazole or posaconazole breakpoints for resistance. *C. glabrata* ATCC 90030 was incorporated into each set of experiments as quality control.

**Adherence assays.** Adherence was assessed using Hep-2 (American Type Culture Collection ATCC CCL23) epithelial cells. Yeast cells in exponential phase were added to confluent Hep-2 cells in RPMI 1640 medium without serum (Sigma R1383) at various multiplicities of infection (MOI). Contact between Hep-2 and yeast cells was initiated by brief (1 to 2 min) centrifugation (500 x *g*)[67]. After incubation for 1 hour, nonadherent cells were removed by five washes in PBS. Adherent cells were recovered by lysis of the monolayer using 0.5 mL of 0.1% triton (Fisher Scientific BP151-100), 0.5 % SDS (Fisher Scientific BP1311-200), 10 mM EDTA in PBS. Adherent yeast cells were scraped off the well, and serial dilutions were made in distilled water, which were then cultured on YPD agar plates for colony count enumeration. Percentage of adherence was calculated by dividing colony forming units (CFUs) of adherent cells over CFU of input cells and multiplied times 100.

**Killing assays by polymorphonuclear cells (neutrophils).** Fresh PMNs were resuspended in RPMI 1640. Opsonized *C. glabrata* cells (50% normal human serum at 37°C for 30 minutes) were incubated with $10^6$ PMNs at an MOI of 50:1 (neutrophil:yeast) in 1 mL of RPMI 1640 medium containing 5% human serum at 37° C[20]. After 2-h of incubation, PMNs were lysed with sterile water, serial 10-fold dilutions were made, and colony counts were enumerated. Fungicidal activity was calculated as the percent of survival of *C. glabrata* after 2-h incubation with neutrophils. Experiments were performed in triplicate and repeated at least twice.

**Paraquat killing.** Yeast was grown overnight in Synthetic Complete Medium (SC medium) at 30 °C with 250 rpm shaking. Cells were washed twice with 1XPBS and adjusted to $5 \times 10^3$/mL. Paraquat (Fisher Scientific USPST740AS) was added to desired concentrations. After 1 hour incubation at 37 °C with 750 rpm shaking, serial 10-fold dilutions were made, and colony counts enumerated after 48 h incubation at 30°C.

**Mouse model of disseminated candidiasis.** CD1 mice (purchased from Envigo, 4–6 weeks old, 1:1 male:female) were maintained in a specific-pathogen-free (SPF) environment at an American Association for the Accreditation of Laboratory Animal Care-accredited animal facility at the University of Pittsburgh. All mice were co-housed at 5 mice maximum in each cage. Animals were assigned randomly to experimental groups. The mice were infected via tail vein injection with $1.5 \times 10^8$ or $1 \times 10^7$ CFU *C. glabrata* strains (mortality and tissue burden determinations, respectively). For mortality studies, mice were followed for 21 days; none showed signs of illness. For tissue burden studies, mice were euthanized and livers, kidneys and spleens harvested on days 1, 3, and 7. Organs were homogenized and dilutions

plated for total CFU counts on YPD. Colonies were counted after 48–72 h of incubation at 30 °C.

## Spiking of sterile blood culture bottles

Stock of strain J1 was prepared from the colony originally isolated from the positive blood culture and stored at −80 °C. We streaked J1 to isolation from stock on SD agar overnight at 30 °C. A single colony was selected and a 100 CFU/mL suspension was prepared in saline. We inoculated 50 μL suspension into a sterile culture bottle (Fisher Scientific BD BACTEC Plus Aerobic medium BD442023) that contained 5 mL blood (-1 CFU/mL), and incubated at 30 °C with shaking at 225 rpm for 48 h. Ten microliters of culture were streaked on an SDA plate, and incubated at 30 °C for 48 hrs. Single strains were isolated from 10 randomly chosen individual colonies, and incubated in 10 mL YPD at 30 °C overnight for gDNA extraction.

## Statistical analysis

MICs and CFU/mL were log-transformed prior to statistical analysis. Data are presented as means and standard error for symmetric data and as medians and interquartile ranges (IQR) for asymmetric data. Statistical analyses were performed using GraphPad Prism version 9.1.2. Student t-test or Mann-Whitney U tests were used for comparisons of 2 groups, and one-way ANOVA or Kruskal-Wallis for comparisons of >2 groups. Survival curves were calculated according to the Kaplan-Meier method using Prism and compared using Newman-Keuls analysis. For all analyses, $p < 0.05$ (two-tailed) was considered significant.

## Reporting summary

Further information on research design is available in the Nature Portfolio Reporting Summary linked to this article.

## Data availability

The datasets generated in this study are included in the article, the Supplementary Information, Reporting Summary and Supplementary Data 1 and 2. The whole genome sequence data generated have been deposited in the NCBI database under accession number PRJNA944162.

## Code availability

All software and programs used in this study and the custom nextflow pipeline script are publicly available at https://doi.org/10.6084/m9.figshare.23596599.

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

## Acknowledgements

This project was supported by NIH grants R21AI160098 (M.H.N.), R21AI152018 (C.J.C.), and 1U54GH009824 and 1R01AI170111-01 (C.L.D.), and U.S. Department of Veterans Affairs Merit grant 1I01BX001955 (C.J.C.). This project was supported in part by the University of Pittsburgh Center for Research Computing, RRID:SCR_022735, through the resources provided. Specifically, the work used the HTC cluster, which is supported by NIH award number S10OD028483. We acknowledge the assistance of Fanping Mu and the entire CRC staff. We are grateful to Mohammed Khalfan from New York University for the publicly available Nextflow variant calling pipeline, which served as the foundation to build our *Candida* variant calling pipeline.

## Author contributions

H.B and G.F: performed whole-genome sequence data analyses; assisted with drafting, editing, and revising manuscript; and prepared genomics tables and figures. S.C.: performed phenotypic characterizations of strains, conducted RT-PCR, and other in vitro and mouse experiments; interpreted data, drafted and revised manuscript; and formatted tables and figures. C.L.D., D.K., and J.L.E.: carried out whole-genome sequencing and accompanying data analyses, and assisted with editing manuscript. E.D., K.M., G.L., B.H., and A.N. collected blood cultures, selected strains from blood culture bottles, species identification of all strains, conducted experiments in conjunction with S.C.; C.J.C. and M.H.N.: study conception and design; directed experiments and data analyses; and redrafted, edited and revised manuscript. They contributed equally to this work.

## Competing interests

The authors declare no competing interests.
