## [Peer Review File · Nature Communications]

Genotypic diversity and unrecognized antifungal resistance among populations of *Candida glabrata* from positive blood culturesReviewers' Comments:

Reviewer #1:

Remarks to the Author:

This manuscript describes an extensive genomic and phenotypic analysis of *C. glabrata* strains isolated from the blood and tissues of patients with invasive infections. The authors find that when multiple colonies from the same blood culture are analyzed, the strains show significant diversity including antifungal resistance. This diversity has substantial clinical implications. Overall, the manuscript is interesting and well-written. Specific comments:

- 1) Fig. 4. This figure needs a title on the y-axis.
- 2) Fig. 5D also needs a y-axis title.
- 3) In the text (line 237), it is stated that the mice were inoculated with $1e7$ cells of each strain of *C. glabrata*, whereas in the Methods section and the legend for Fig. 6, it is stated that the inoculum was $1.5e8$.
- 4) In the paragraph beginning on line 371, the authors should note that another possible reason for their finding that the SCV strain had attenuated virulence in mice, yet still caused disease in the patient is that unlike *C. albicans*, *C. glabrata* is a very weak pathogen in mice, in contrast to humans, suggesting that the pathogenesis of *C. glabrata* infection in mice is different from that in humans.
- 5) As a minor point, the identity of the wild-type strain in which the PDR1 mutant allele was inserted should be provided in the Methods section.

Reviewer #2:

Remarks to the Author:

This is in many aspects an interesting manuscript. The authors show, via whole genome sequencing of several colonies from a blood culture, that *C. glabrata* yeasts isolated from the bloodstream are not genetically uniform. One conclusion may be that they are not derived from a single infection site/event, or that mutations accumulate during the active infection. They also found petite (small colony variant) strains in the blood of one patient, proving recent observations that these may be present, but overlooked.

This information is of some relevance to the field, as it challenges the assumption, held by many, that a single colony represents the whole infecting population in BSI.

There are some aspects that the authors may want to explain in more depth:

Line 392 -- The authors say that it is unlikely that the detected mutations appeared within the blood culture itself. They compare the number of mutations of a single strain grown in sterile blood with the average number of mutations between strains from a single patient. Here, I think a deeper analysis may be required.

First, if I understand correctly, the intra-patient number of mutations is comparable among the strains? Even if the colonies originated from several distinct clones, would we not expect some colonies to come from the same original bloodstream yeast? In that case, would there be more clustering within the intra-patient samples? Can the authors provide an estimate of how many yeasts would have to be in the original patient sample (before incubation) to not see colonies from the same strain with their sample size of 10?

Also, for a better comparability, could the same experiment be performed with another strain than J1? Especially the J group, with its SCVs, seems not the most representative basis for comparison of mutation rates of all patient samples? The information given by the authors is a bit sparse (e.g. line 255), and I would like some more quantitative data to support this major conclusion for the paper.

Also, the claim that the current paradigm assumes single-source infections and that laboratory testing requires only single colonies to be tested needs some critical evaluation. To my knowledge, e.g.

EUCAST requires 5 colonies (distinct or not) to be tested. The presence of strains with different resistance profiles from sterile body sites including blood cultures has been shown e.g. by Knoll et al (PMID 35537592). So this paradigm may be true for many, but not all clinicians and scientists, and this may be discussed in some more detail.

Some minor points:

Line 474 -- alignment-from-vcf does not seem to exist any more on github.com/Bahler-Lab/

Table 3 -- Italicize gene names for patient I, remove italics for systematic identifiers for patients L, EF, J, K, ST

Figure 5C -- FACs -> FACS

Reviewer #3:

Remarks to the Author:

The manuscript "Genotypic diversity and unrecognized antifungal resistance among populations of *Candida glabrata* from positive blood cultures" by Badrane and colleagues provides an unprecedented set of data that will contribute to our knowledge of antifungal resistance acquisition in the pathogenic species *Candida glabrata*. Provided that the number of lethal systemic infections by antifungal-resistant strains belonging to this species, the improvement of our knowledge of how it develops is fundamental.

The study confirms several of the previously observed genomic characteristics associated with some features relevant to the pathogenic and lethal characteristics of *C. glabrata* strains:

- links between petit phenotype (small colonies, associated with mitochondrial dysfunctions due to mutations in mtDNA or in nuclear genes involved in oxidative phosphorylation) and resistance to antifungals – e.g. doi: 10.1128/mBio.01128-21; doi: 10.1111/j.1567-1364.2012.00821.x

- enrichment of variations in genes coding for adhesin or adhesin-like proteins in clinical *C. glabrata* strains – e.g. doi: 10.1016/j.cmi.2017.03.014; doi: 10.1128/spectrum.01827-22

The great novelty of the study by Badrane and colleagues, however, resides in the set of strains studied: differently from previous studies, comparing strains isolated from different patients, this study compares multiple strains isolated from the same patient. This, also combined with the in vitro test that the authors performed (culturing strains in sterile blood to assess the insurgence of petit phenotype), provides fundamental insights into the dynamics and timing of antifungal resistance acquisition.

The methodology is multifaceted and appropriate for the type of investigation, and the interpretation and conclusions are properly supported by the gathered data.

I have only a few minor comments, please see the details below.

Overall, the study represents a very important piece of work from many viewpoints including clinical microbiology and evolution and poses a solid basis for the research of proper clinical therapies against antifungal-resistant infections.

Minor comments:

It would be very useful to provide a supplementary table summarizing all the features of the isolated and investigated strains. For instance, the correspondence between the colony size and the nuclear and mitochondrial sequences clustering. This information comes out only at the time of phenotype characterization.

L91: The meaning of "index strain" should be provided here, not only in the methods section.

Figure 2B: why just part of the intra-patient SNP distances are shown? e.g. strains from patient J compared only to strains from patients K, P, ST and not to patients C, D, I, L, EF, and H? Is it because only strains from the same clade were compared among each other?

Figure 2C: The font size should be increased, especially for the text in the clusters' panels, which is

barely readable.

Figure 4: Please indicate the scale of the y legend (how are the RT-PCR results quantified?).

L184: is "institution" the right word?

Chromosomal rearrangements, rather than chromosomal aneuploidies or polyploidies have been found in several clinical *Candida glabrata* strains -see doi: 10.1128/spectrum.01827-22; doi: 10.1016/j.cub.2017.11.027). The authors should check if the strains isolated in their study show the same characteristics.

In addition, recent studies have already identified a connection between changes in genes coding for adhesin and adhesin-like proteins and the phenotype (pathogenicity and antifungal resistance (doi: 10.1016/j.cmi.2017.03.014; doi: 10.1128/spectrum.01827-22).

Why "Candida", when referring to the entire genus is not in italics?

Please note that mouse experiments included equal numbers of male and female mice. This has been explicitly described in the Methods section (lines 615-616). There were no differences in survival or tissue burdens among male and female mice during hematogenous disseminated candidiasis (please refer to the Source data file (Excel work sheet labeled Fig. 6). In keeping with Nature's guidance on reporting on sex and gender, this information has been added to the paper.

Reviewer #1

1) Fig. 4. This figure needs a title on the y-axis.

- The y-axis title has now been added to Figure 4. In addition, Fig. 6 legend further defines the Y-axis "Y-axis represents fold difference in *CDR1* or *PDR1* expression in specific L strains relative to L1, normalized to 18S"

2) Fig. 5D also needs a y-axis title.

- The Y-axis has now been added to Figure 5D "The Y-axis presents cell numbers, as the percentage of the total number of cells."

3) In the text (line 237), it is stated that the mice were inoculated with 1e7 cells of each strain of *C. glabrata*, whereas in the Methods section and the legend for Fig. 6, it is stated that the inoculum was 1.5e8.

- For the mortality study, mice were infected with 1.5×10^8 CFU. For the tissue burden study, they were infected with 1×10^7 CFU. We have clarified the inocula used in the Results (lines 238-239) and Methods sections (lines 616-617).

4) In the paragraph beginning on line 371, the authors should note that another possible reason for their finding that the SCV strain had attenuated virulence in mice, yet still caused disease in the patient is that unlike *C. albicans*, *C. glabrata* is a very weak pathogen in mice, in contrast to humans, suggesting that the pathogenesis of *C. glabrata* infection in mice is different from that in humans.

- Thank you for making this important point. The following sentence has been added: "*Moreover, C. glabrata is a relatively weak pathogen in mice, and limited virulence in mouse infection models does not necessarily correspond to lack of pathogenicity in humans*" (lines 401-403)

5) As a minor point, the identity of the wild-type strain in which the PDR1 mutant allele was inserted should be provided in the Methods section.

- Please note that we now indicate in the Methods sections that gene manipulation was performed in 2 strains: "*For PDR1 replacement, the complete PDR1 ORF flanked by 500 bp was amplified by PCR from genomic DNA of BG2 (with wild-type PDR1) and L4 (with G346C PDR1)*".

Reviewer #2:

Line 392 -- The authors say that it is unlikely that the detected mutations appeared within the blood culture itself. They compare the number of mutations of a single strain grown in sterile blood with the average number of mutations between strains from a single patient. Here, I think a deeper analysis may be required. First, if I understand correctly, the intra-patient number of mutations is comparable among the strains? Even if the colonies originated from several distinct clones, would we not expect some colonies to come from the same original bloodstream yeast? In that case, would there be more clustering within the intra-patient samples? Can the authors provide an estimate of how many yeasts would have to be in the original patient sample (before incubation) to not see colonies from the same strain with their sample size of 10? Also, for a better comparability, could the same experiment be performed with another strain than J1? Especially the J group, with its SCVs, seems not the most representative basis for comparison of mutation rates of all patient samples? The information given by the authors is a bit sparse (e.g. line 255), and I would like some more quantitative data to support this major conclusion for the paper.

- As requested by the Reviewer, we have added extensive new data and text in the Results (lines 256-267) and as new worksheet in Supplemental Table 2A (under "J1-J10 vs Spiked BCx") and a new Supplemental Table 5A. Specifically, the following has been added:

"Strains obtained from spiked blood culture demonstrated 2 ± 0.08 -fold fewer nucleotide differences than strains J1-J10 ($p < 0.0001$ for both nuclear and mitochondrial genomes). (line 255)

"We analyzed variants associated with non-synonymous SNPs/indels that discriminated within-patient strains J1-J10 and those that discriminated strains recovered from the spiked blood culture bottle. There were 115 non-synonymous variant-containing genes that discriminated among J1-10 that were not identified in the spiked culture bottle [Supplemental Figure 3A]. GO term analysis of these variants revealed genes enriched in processes linked to mitochondrial functions involved in respiratory electron transfer and ATP synthesis [Supplemental Figure 3B]. There were 69 non-synonymous variant-containing genes that discriminated among both J1-J10 and strains from the spiked bottle; genes encoding cell wall proteins involved in adhesion were over-represented. Only 20 non-synonymous variant-containing genes were present uniquely in strains from the spiked sample; no biologic process or function was significantly over-represented among these genes." (lines 256-267)

The author raises a good point that with our study design we cannot say definitively that all of the diversity we observed emerged in the patient, rather than within the blood culture itself. We have now revised our Discussion to acknowledge this point:

“Our study was designed to collect C. glabrata as blood cultures were being processed according to standard clinical microbiology lab practices. As such, a strength of the study is that results reflected C. glabrata diversity that might be unrecognized by clinicians. We acknowledge that we cannot definitively determine when specific mutations may have arisen. However, our data suggest that most within-patient C. glabrata diversity emerged in vivo, rather than in vitro. Indeed, there were twice as many nucleotide differences among strains from patient J’s blood culture than there were among strains from a culture bottle spiked with strain J1 ($p < 0.0001$). Furthermore, SCV strains did not emerge in the spiked culture. It is unclear whether BSI diversity may have stemmed from one-time inoculation of the blood with a mixed population of strains from sites of colonization such as the GI tract, serial inoculation of strains, and/or mutations emerging in the bloodstream prior to collection of cultures. We believe the majority of diversity that arose in the host was likely to do so at colonization sites since Candida population sizes are typically greater and durations of infection are longer during colonization than they are for candidemia” (lines 410-424)

Likewise, we acknowledge that our study design does not allow us to definitively say that each strain in all patients is unique and diverse. Therefore, we have removed all text throughout the manuscript that implied such an interpretation of the data. Nevertheless, the constellation of genetic differences we observe in each patient, the greater number of nucleotide differences within a patient’s blood cultures than in a spiked sample, and the presence of phenotypes that correlate with genotypes supports our conclusions. Since we cannot conclusively determine how many of the 10 strains picked from a given patient were truly distinct genetically, it is not possible to estimate the number of yeasts that might be present in the original patient sample.

The rationale for including J1 in the spiking study was that the SCV phenotype would allow us to determine if an important phenotype seen in patient J’s blood culture could be replicated by culturing *in vitro*. The fact that SCVs did not arise when J1 was spiked into the blood culture bottle suggests that phenotypic and genotypic diversity in patient J’s blood culture emerged *in vivo*.

Also, the claim that the current paradigm assumes single-source infections and that laboratory testing requires only single colonies to be tested needs some critical evaluation. To my knowledge, e.g. EUCAST requires 5 colonies (distinct or not) to be tested. The presence of strains with different resistance profiles from sterile body sites including blood cultures has been shown e.g. by Knoll et al (PMID 35537592). So this paradigm may be true for many, but not all clinicians and scientists, and this may be discussed in some more detail.

We thank the reviewer for this comment. The publication by Knoll was cited. We have now added the EUCAST reference recommending testing 5 colonies. We have also revised the discussion to make this point, and removed a sentence that said susceptibility testing was based on a single strain from a single colony. Specifically, in the Introduction and Discussion, the following sentences were removed “Standard microbiology laboratory protocol in processing positive clinical cultures is to test single microbial strains from morphologically distinct colonies. (lines 51-53) and “If validated in other studies, results have potential implications for medical and clinical microbiology practices, and for understanding emergence of *C. glabrata* antifungal resistance, treatment responses, pathogenesis and adaptation. (lines 288-291). We added the following sentence “The European Committee on Antimicrobial Susceptibility Testing currently recommends measuring antifungal MICs against suspensions of up to 5 representative colonies ⁴⁷, but this practice has not been endorsed by the Clinical and Laboratory Standards Institute ⁴⁸.” (lines 460-463)

Some minor points:

Line 474 -- alignment-from-vcf does not seem to exist any more on github.com/Bahler-Lab/

- Methods for alignment have been update with “*a whole genome multiple sequence alignment (MSA) was extracted from the multi-sample VCF file using VCF2MSA python script (<https://github.com/tkchafin/vcf2msa.py>).*” (lines 498-500)

Table 3 -- Italicize gene names for patient I, remove italics for systematic identifiers for patients L, EF, J, K, ST

- We thank the reviewer for drawing attention to these errors, which have been corrected

Figure 5C -- FACs -> FACS

- This change has been made.

Reviewer #3:

Overall, the study represents a very important piece of work from many viewpoints including clinical microbiology and evolution and poses a solid basis for the research of proper clinical therapies against antifungal-resistant infections.

- We thank the reviewer for his/her positive comments. All references cited by the reviewer are included in our manuscript.

Minor comments:

It would be very useful to provide a supplementary table summarizing all the features of the isolated and investigated strains. For instance, the correspondence between the colony size and the nuclear and mitochondrial

sequences clustering. This information comes out only at the time of phenotype characterization.

- We thank the reviewer for this suggestion. An additional table summarizing the genotypes and phenotypes of strains from patients J and L has been added to supplemental material.

L91: The meaning of “index strain” should be provided here, not only in the methods section.

- The definition of “index strain” is now provided in the Introduction (lines 64-66) and Results sections (line 78).

Figure 2B: why just part of the intra-patient SNP distances are shown? e.g. strains from patient J compared only to strains from patients K, P, ST and not to patients C, D, I, L, EF, and H? Is it because only strains from the same clade were compared among each other?

- Fig 2B shows SNP distances between strains from all 10 patients. These 94 strains represent 4 STs from 3 distinct clades.

Figure 2C: The font size should be increased, especially for the text in the clusters’ panels, which is barely readable.

- Fig 2C has been revised so that the font size is now more readable

Figure 4: Please indicate the scale of the y legend (how are the RT-PCR results quantified?).

- We have now split Figure 4 into Figure 4a (*PDR1*) and 4b (*CDR1*). The scale of the Y-axis is now clearly denoted

L184: is “institution” the right word?

- “We thank the reviewer for this question. We have edited the sentence to “*Upon diagnosis of fluconazole-susceptible C. glabrata BSI, patient L was treated with IV fluconazole. Blood cultures ~48 hours later remained positive for C. glabrata, which was now identified by the clinical microbiology laboratory as azole-resistant*” (lines 159-162)

Chromosomal rearrangements, rather than chromosomal aneuploidies or polyploidies have been found in several clinical *Candida glabrata* strains -see doi: 10.1128/spectrum.01827-22; doi: 10.1016/j.cub.2017.11.027). The authors should check if the strains isolated in their study show the same characteristics.

- We thank the reviewer for this suggestion. Chromosomal rearrangement data have now been added to the Results (lines 132-138), Discussion (line 274; and

lines 340-343) and Methods section (lines 521-522). We also added Supplementary Fig. 2A. The references provided are cited in the revised MS.

“To look for structural variants, genomes of within-patient strains were aligned using progressiveMauve, then visualized in Mauve Alignment Viewer. Graphics did not reveal aneuploidies, but provided evidence of chromosomal rearrangements [Supplementary Figure 2A]. We used similarity dot plots of pairwise genome comparisons (D-Genies) to confirm chromosomal rearrangements in strains from 8 patients (patients K and ST excepted) [Supplementary Figure 2B].

*“In contrast, we observed chromosomal rearrangements among strains from 8 of 10 patients. Chromosomal rearrangements are frequent and highly dynamic in *C. glabrata*^{8,35}, and they are postulated to represent adaptive mechanisms for survival in the host³⁶.”*

In addition, recent studies have already identified a connection between changes in genes coding for adhesin and adhesin-like proteins and the phenotype (pathogenicity and antifungal resistance (doi: 10.1016/j.cmi.2017.03.014; doi:10.1128/spectrum.01827-22)).

- Thank you. We have now added doi:10.1128/spectrum.01827-22 (new reference #36). Reference doi: 10.1016/j.cmi.2017.03.014 has already been cited (reference #8)

Why "*Candida*", when referring to the entire genus is not in italics?

- *Candida* is now italicized when used to refer to the entire genus.

Reviewers' Comments:

Reviewer #2:

Remarks to the Author:

All my comments have been adequately addressed, thank you and good luck with your interesting work!

Reviewer #3:

Remarks to the Author:

The authors have properly addressed my concerns.

I was already enthusiastic and impressed by the work, and now I am happy to endorse the manuscript for publication.